# Effects of Tributyltin-Contaminated Aquatic Environments and Remediated Water on Early Development of Sea Urchin (*Hemisentrotus pulcherrimus*)

**DOI:** 10.3390/ani13193078

**Published:** 2023-10-01

**Authors:** Hee-Chan Choi, Ju-Wook Lee, Un-Ki Hwang, Ha-Jeong Jeon, Sung-Yong Oh, Chul-Won Kim, Han-Seung Kang

**Affiliations:** 1Marine Environment Impact Assessment Center, National Institute of Fisheries Science, Busan 46083, Republic of Korea; gmlckschl82@korea.kr; 2West Sea Fisheries Research Institute, National Institute of Fisheries Science, Incheon 22383, Republic of Korea; leejuwook84@gmail.com; 3Tidal Flat Research Center, West Sea Fisheries Research Institute, National Institute of Fisheries Science, Gunsan 54001, Republic of Korea; 4Department of Marine Environment, MS BioLab, Daejeon 34576, Republic of Korea; 5Marine Biotechnology & Bioresource Research Department, Korea Institute of Ocean Science & Technology, Busan 49111, Republic of Korea; 6Department of Aquaculture, Korea National College of Agriculture and Fisheries, Kongjwipatjwi-ro 1515, Wansan-gu, Jeonju 54874, Republic of Korea

**Keywords:** *Hemicentrotus pulcherrimus*, tributyltin, gametotoxicity, embryotoxicity, development impairment

## Abstract

**Simple Summary:**

In this study, the toxicity of tributyltin (TBT) was investigated. Sea urchins (*Hemicentrotus pulcherrimus*) were used as the experimental animals, and the embryonic toxicity of TBT was investigated by observing the fertilization and embryonic development rates of germ cells exposed to TBT. The fertilization rate exhibited decreasing trends as the concentration of TBT increased. Embryo development was delayed as the TBT concentration increased. Embryos whose development was delayed following TBT exposure progressed well when shifted to fresh media without TBT. The results showed that TBT had a negative effect on embryonic development. Embryonic development was restored on removal of TBT exposure. This result has important implications from the perspective of restoring polluted ecosystems.

**Abstract:**

In this study, gametotoxicity and embryotoxicity experiments were performed using *Hemicentrotus pulcherrimus* to investigate the toxic effects of tributyltin (TBT). The effects of TBT on fertilization and embryogenesis were assessed at various concentrations (0, 0.02, 0.05, 0.09, 0.16, 0.43, 0.73, 4.68, and 9.22 ppb). The fertilization rates decreased in a concentration-dependent manner, with significant reduction following treatment with TBT at 0.05 ppb. Embryos exhibited developmental impairment after TBT exposure at each tested concentration. The frequency of developmental inhibition delay that treatment with TBT delayed embryonic development in a dose-dependent manner, with 100% of embryos exhibiting developmental impairment at 4.68 ppb. During developmental recovery tests, embryos cultured in fresh media without TBT showed advanced embryonic development. Although the observed normal development after transferring the developmentally delayed embryos to fresh media without TBT offers prospects for the restoration of contaminated environments, embryonic development remained incomplete. These results suggest that TBT adversely affects the early embryonic development of *H. pulcherrimus*.

## 1. Introduction

Endocrine-disrupting chemicals (EDCs) interrupt the endocrine system by increasing or blocking the synthesis, release, and activity of natural hormones, thereby disrupting the physiological effects of endogenous hormones [1,2]. EDCs are divided into natural substances, such as sex hormones, phytoestrogens, and mycoestrogens, and chemical substances, such as organotins (tributyltin), bisphenol A, phthalates, dioxins, pesticides, organophosphates, polychlorinated biphenyls (PCB), diethylstilbestrol (DES), and polybrominated diphenyl ethers (PBDEs) [3,4,5].

Tributyltin (Tributyltin oxide, TBT) is an organotin compound with the chemical formula C_24_H_54_OSn_2_ that is used as an antifouling agent in paints to prevent the settlement and growth of aquatic organisms on submerged structures and aquaculture nets [6]. TBT is a highly toxic organic compound that has endocrine disrupting activity in aquatic faunae and has been demonstrated to be toxic to the embryos and larval stages of numerous aquatic animals, even at environmentally relevant concentrations [7,8,9]. In addition, TBT is a potent biocide that is widely used in several industrial processes and is considered a persistent organic pollutant [10,11]. Although the use of TBT has been regulated in many countries since the 1980s, it has been detected in water, sediments, and biological tissues in several countries [12,13]. Notably, it is present at high levels in harbors, coastal areas, and industrial complexes [14,15,16]. Low concentrations (in the order of ng/L, ppt) of TBT can cause sublethal effects in marine organisms [16,17,18]. It may have harmful effects on embryonic development, sex ratio, sexual behavior, and gonadal development and has been implicated in imposex development [6,19,20,21]. In particular, it has been demonstrated to be toxic to the embryonic and larval stages of numerous aquatic organisms, even at environmentally relevant concentrations [9]. Imposex was induced at very low concentrations, such as 1 ng/L, in a *Nucella lapillus* population; concentrations above 2 ng/L inhibited proper calcification of oyster (*Crassostrea gigas*), whereas concentrations above 20 ng/L inhibited larval growth of the organism [22,23]. Moreover, TBT induces adipocyte differentiation, ectopic adipocyte formation, and lipid accumulation [24,25]. Furthermore, it induces the enlargement of lipid droplets in the testes, exerting negative effects on reproduction [26].

The sea urchin (*Hemicentrotus pulcherrimus*) embryo is an intact developmental system that undergoes events comparable to those of vertebrates, including mammals. The assessment of sea urchin embryogenesis and teratogenesis is convenient because sea urchins can rapidly provide information on developmental toxicants. In recent decades, sea urchin larval development has been studied and used to monitor pollutants in marine environments [27,28,29]. Two main life stages, larvae and adults, have mostly been studied and used for testing purposes [28]. The early-life stages of sea urchins are particularly sensitive to metals [27,29]; hence, sea urchins were used as model organisms in our study because of their high sensitivity to chemical compounds, such as TBT. We examined the effects of TBT on the rates of fertilization, survival, and normal development post-fertilization in *H. pulcherrimus* gametes and larvae. We also observed the recovery of TBT-exposed organisms in clean areas away from the TBT exposure.

The results of the present study suggest that the level of toxicity affecting early embryos depends on TBT concentration. In addition, organisms exposed to TBT contamination can indicate the quality of a restored ecosystem.

## 2. Materials and Methods

### 2.1. Experimental Sea Urchin and Study Conditions

Sea urchins (*H. pulcherrimus*), which belong to the marine invertebrate class Echinodermata, were used in this study. They inhabit the rocky areas along the entire Korean coast. Adult *H. pulcherrimus* can be easily collected at any time because they have different spawning periods that depend on the water temperature across the intertidal zones. Sea urchins were collected from the intertidal zones of the beaches in Gyeokpo-ri, Byeonsan-myeon, Buan-gun, and Jeollabuk-do from March to April 2021, corresponding to the main spawning period. The collected sea urchins were acclimatized in a laboratory aquarium with a water temperature of 9 ± 1 °C for one week prior to the experiments. The seawater was filtered (using a membrane filter with a pore size of 0.45 μm) and sterilized before use. Seawater was used to remove protozoa and other foreign matter from sea urchin surfaces. Eighteen sea urchins, at least 3.5 cm in diameter, were used for spawning. To obtain mature eggs and sperm, a beaker was filled with seawater to sufficiently soak the genital pore, and 1 mL of 0.5 M KCl solution was injected into the coelomic cavity of the sea urchin using a syringe. The eggs and sperm were collected 30 min after spawning. Sperm and eggs were washed once and thrice, respectively, in seawater before use in the bioassay [30].

### 2.2. Experimental Design

TBT (C_24_H_54_OSn_2_, CAS No. 56-35-9) was purchased from Sigma-Aldrich (St. Louis, MO, USA) and dissolved in absolute ethanol. Fertilized eggs and embryos were obtained from at least three different male–female pairs for each bioassay. The sperm was exposed to varying nominal concentrations of TBT (0, 0.01, 0.05, 0.1, 0.25, 0.5, 1, 5, and 10 ppb in sterilized sea water) for 30 min, after which the ova were injected [30]. Fertilization rates were determined by checking the formation of the fertilization membrane after 10 min. Embryos were transferred to capped tubes, fixed with formalin solution (3%), and observed under an optical microscope (Figure 1). To examine the normal embryogenesis rate, specimens were fixed with formalin solution (3%) 64 h after fertilization. They were then observed using an optical microscope and classified into normal and abnormal (small and deformed) pluteus (Figure 1). The experiment was replicated three times in the same medium, and 100 or more embryos were exposed to the test solution and counted three times to determine the ratio of normal to fixed embryos. The embryos were cultured in an incubator (LMI-3004PL, DAIHAN Lab Tech, Korea) at 16 ± 0.5 °C for 64 h [30]. The pH of the culture fluid was maintained at 7.8–8.2. The detailed culture conditions are listed in Table 1. Using a preliminary test for optimal sperm addition, the sperm was diluted 2000–2500 times, and 1500–2000 fertilized eggs were added to 1 mL of test water used for the culturing process.

For the recovery test, the test samples were used as morula-stage embryos 4 h after vitro fertilization. The test medium was not replaced in the control culture. However, the test medium was replaced with fresh medium in the cultured embryos exposed to TBT at 5 ppb, and the embryos were cultured for an additional 24 or 64 h at 16 °C in an incubator.

### 2.3. TBT Concentration Analysis

Analysis of TBT concentration in seawater was performed as described previously [16]. Unfiltered 500 mL samples were extracted twice by shaking for 1 h using 50 mL of 0.1% tropolone (Merck, Hohenbrum, Germany) and methylene chloride (Merck, Darmstadt, Germany), along with the addition of 5 mL 50% HCl (Merck, Darmstadt, Germany) in separator funnels. Triphenyltin chloride (Kanto, Tokyo, Japan) was used as the surrogate standard before extraction. The total organic extract (0.5 mL) was concentrated and transferred to hexane (Merck, Darmstadt, Germany). The extract was hexylated with 1 mL Grignard reagent and n-hexylmagnesium bromide (TCI, Tokyo, Japan). The remaining Grignard reagent was removed using 5 mL 1NH_2_SO_4_ (Merck, Hohenbrunn, Germany). The organic fraction was decanted, and the aqueous fraction was extracted with hexane (5 mL). The combined organic phase was concentrated using a Turbo Vap LV (Caliper Life Science Inc., Hopkinton, MA, USA) and cleaned by passing through a Florisil column (60–100 mesh, reagent grade, Sigma-Aldrich, St. Louis, MO, USA). The eluents were concentrated to 0.5 mL under N_2_. Finally, tetrabutyltin (Sigma-Aldrich, St. Louis, USA) was added to the concentrated eluents as an internal standard.

### 2.4. Statistical Analysis

Data were analyzed via one-way analysis of variance (ANOVA, Fisher’s PLSD test) using Graph Pad Prism ver. 6 (Graph Pad Software, La Jolla, CA, USA). Fisher’s exact test was used to examine the significance of the correlation coefficients. Significance was accepted at ** *p* < 0.01, * *p* < 0.05.

## 3. Results and Discussion

To investigate the effects of TBT on gametotoxicity and embryotoxicity, sperm and fertilized eggs of *H. pulcherrimus* were treated with different doses of TBT dissolved in filtered and sterilized seawater for 30 min and 64 h, respectively. TBT concentrations in sea water at the end of the 64-h exposure were lower than nominal concentrations, except for medium containing 0.01 or 0.05 ppb of TBT. The average TBT concentrations in the experimental seawater of the 0.01, 0.05, 0.1, 0.25, 0.5, 1, 5, and 10 ppb groups were 0.02, 0.05, 0.09, 0.16, 0.43, 0.73, 4.68, and 9.22 ppb, respectively. During the recovery test, the average TBT concentration in the seawater medium at the end of 24 and 64 h was 5 ppb. The concentration of TBT in the treatment groups were 4.87 and 4.73 ppb. The gametotoxicity and embryotoxicity experiments revealed that upon exposure to TBT, fertilized egg samples exhibited certain abnormalities, including occurrence of unfertilized eggs and ruptured fertilized eggs (Figure 1).

Following TBT treatment, fertilization rates decreased when compared with those in the control, unexposed to TBT (Figure 2). The average fertilization rate in the control was 96.67%. However, the average percentages of normally fertilized eggs at each concentration (0, 0.02, 0.05, 0.09, 0.16, 0.43, 0.73, 4.68, and 9.22 ppb) of TBT were 96.67, 91.33, 78.33, 72.67, 72.67, 74.10, 81.00, 80.33, and 70.33%, respectively. Fertilization rates following treatment with 0.05 ppb were significantly lower than those of the control. For sperm treated with TBT, the lower fertilization rate might indicate acrosin and/or acrosome deficiency, as the alteration of acrosin is consistent with an effect on sperm viability. Acrosin is a constituent of the mammalian sperm acrosome and is active only during acrosomal reactions. The acrosome reaction begins when sperm binds to zona pellucida glycoproteins. Acrosin is a crucial enzyme that maintains the normal form of sperm [31] and facilitates penetration of the egg’s zona pellucida [32]. Therefore, when acrosin is inhibited, sperm cannot bind to or penetrate the zona pellucida, thereby preventing fertilization [32]. TBT is known to decrease acrosin activity [31]; therefore, the low fertilization rate of sea urchin sperm treated with TBT in the present study may indicate acrosin deficiency. Acrosin activity has been shown to decrease in the presence of small amounts TBT in a dose-dependent manner [33]. Furthermore, taurine supplementation has been reported to increase acrosin levels and enhance sperm quality and function in mammals [34].

Our results showed that when exposed to TBT, the normal embryogenesis rate of pluteus larvae gradually decreased in a concentration-dependent manner, and the embryos displayed developmental impairment after treatment at each concentration (Figure 3). The frequencies of normal pluteus larvae following treatment with each concentration (0, 0.02, 0.05, 0.09, 0.16, 0.43, 0.73, 4.68, and 9.22 ppb) of TBT were approximately 83, 57, 57, 57, 42, 35, 43, 0, and 0%, respectively (Figure 3). Another symptom of TBT exposure is developmental impairment, which is implicated in the proportions of abnormal-, morula-, blastula-, and gastrula-stage embryos. Increasing concentrations of TBT could induce developmental impairment, and all embryos were observed to be impaired at a concentration of 4.68 ppb. The frequency of embryos showing an impaired developmental stage increased with increasing TBT concentration. The embryos developed to the pluteus larvae stage during treatment from 0.02 to 0.73 ppb of TBT; whereas the embryos remained at the morula and/or blastula stage when treated with 4.68 to 9.22 ppb of TBT. Similarly, a previous study has reported impairment of embryonic development in the sea urchins upon TBT treatment [35,36].

Organotin TBT, triphenyltin (TPT), and the herbicides 2-methylthio-4-t-butylamino-6-cyclopropylamino-s-triazine (Irgarol 1051), 3-(3,4-dichlorophenyl)-1,1-dimethylurea (Diuron) were evaluated for toxicity in the sea urchin *Lytechinus variegatus*. The results showed that embryotoxicity decreased in the following order: TBT > TPT > Irgarol 1051 > Diuron. Additionally, as the concentration increased, embryonic development was delayed [36]. TBT affects biochemical processes such as the inhibition of mitochondrial oxidative phosphorylation, genotoxicity, DNA damage, ATP synthesis, inhibition of ATPases, and perturbation of calcium homeostasis [36]. Calcium ions play a critical role in TBT-induced apoptosis by disrupting Ca^2+^ homeostasis [36,37]. Alterations in calcium signaling cause developmental toxicity in sea urchin embryos and larvae [35,36]. TBT stimulates plasma membrane permeability because of the distribution of organotin molecules in membrane phospholipids. Furthermore, the TBT-induced decrease in the intracellular compartmentalization of Ca^2+^ may be related primarily to the inhibition of the ATP-driven Ca^2+^ pump [36]. Therefore, developmental impairment may be one of the effects associated with the interference of antifouling compounds with intracellular calcium homeostasis. Exposure of chemicals to germ cells of organisms in aquatic ecosystems are known to have adverse effects on fertilization and embryo development. Embryotoxicity test in oysters (*Crassostrea gigas*) by exposing germ cells to irganol and diuron showed that the fertilization rate decreased and abnormalities in embryo development increased in a concentration-dependent manner [38]. Methoxychlor (1,1,1-trichloro-2,2-bis [p-methoxyphenyl] ethane) is a prohibited chlorinated hydrocarbon developed as a replacement for DDT. Exposure of sea urchin (*Strongylocentrotus purpuratus*) germ cells to methoxychlor resulted in abnormal embryogenesis and inhibition of embryonic development in a concentration-dependent manner [39].

During the recovery test, embryos exposed to 4.87 ppb TBT in the culture medium reached the morula stage. After, being shifted to fresh medium without TBT, the morula-stage embryos progressed to the blastula stage within 24 h, whereas the control embryos reached the gastrula stage at the time point (Figure 4). After culturing for 64 h, approximately 83% of the control embryos had reached the pluteus larva stage, whereas 97% of the embryos that were cultured continuously in 4.73 ppb TBT were in the blastula stage at the corresponding time point. When the medium was replaced with a fresh medium without TBT for the developmental recovery test, 43% and 57% of the embryos were in the blastula and gastrula stages, respectively (Figure 5). During the development recovery test, in which fresh or original medium was applied to the morula-stage embryos that were initially cultured in 4.73 ppb TBT, different results were obtained. Embryos cultured in fresh media without TBT showed advanced embryonic development. However, the embryos did not progress to complete embryonic development. A similar study was conducted using Triclosan (TCS, 2,2,4′-trichloro-2′-hydroxydiphenyl ether) and sea urchin (*Strongylocentrotus nudus*) embryos. TCS is a widely used antibacterial agent. Fertilization rate and embryo development were examined after treating *S. nudus* germ cells with TCS. The results of the study showed a decreased fertilization rate and delayed embryo development in a concentration-dependent manner. Aligning with our result, recovery test of embryos exposed to TCS demonstrated that embryo recovery decreased with increasing exposure time [40]. These results suggest that embryos exposed to pollutants such as TBT at an early stage may not completely recover developmentally, even if the pollutant is removed from the environment.

## 4. Conclusions

In the present study, TBT affected germ cell fertilization and decreased fertilization rate. In addition, it adversely affected embryogenesis and delayed the development of normal embryos. However, embryos exposed to TBT tended to develop normally in the absence of TBT. The results indicate that the restoration and maintenance of clean ecosystems are vital. Terrestrial and marine ecosystems are currently polluted with various environmental pollutants. Environmental pollutants affect the birth and survival of organisms, and birth organisms exhibit and increased frequencies of deformities. Therefore, global citizen should recognize the risk of environmental pollution and the need to restore the health of ecosystems. Ecosystems that recover from environmental pollution can maintain a healthy global ecosystem by providing safe and healthy habitats for various organisms.

## Figures and Tables

**Figure 1 animals-13-03078-f001:**
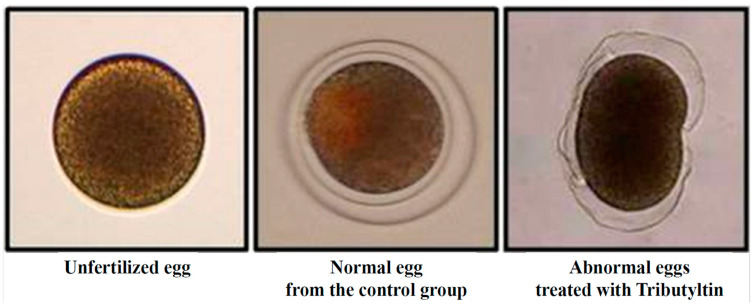
Normal and abnormal forms in fertilized sea urchin eggs (*Hemicentrotus pulcherrimus*).

**Figure 2 animals-13-03078-f002:**
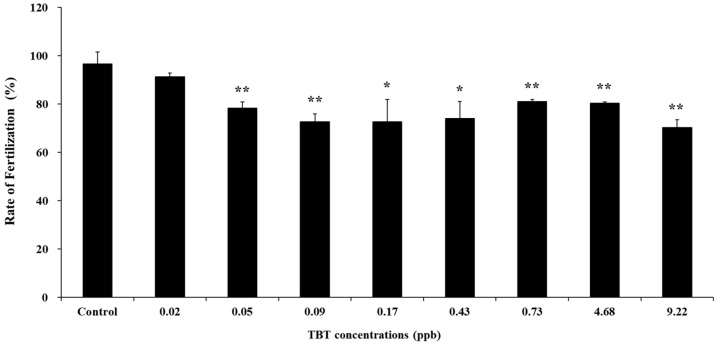
Fertilization rates of sea urchin (*Hemicentrotus pulcherrimus*) sperm exposed to TBT. Vertical bars represent the SD of the mean for three times. * Significantly different from the control by Fisher’s exact test (*p* < 0.05), ** *p* < 0.01.

**Figure 3 animals-13-03078-f003:**
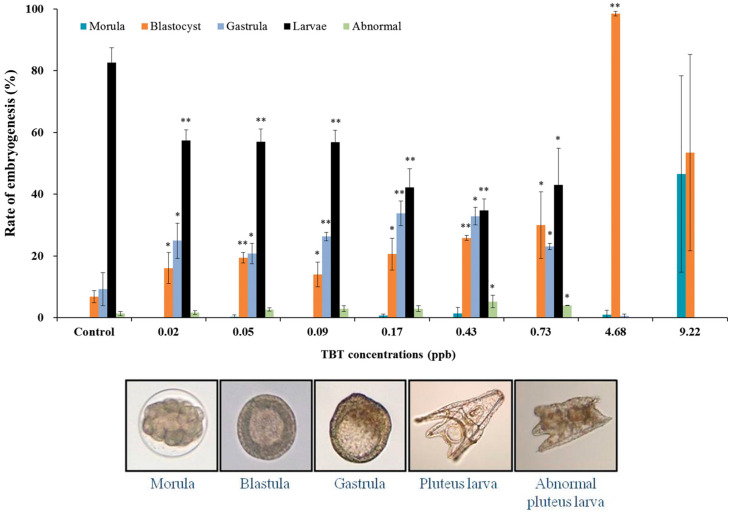
Embryogenesis of sea urchin (*Hemicentrotus pulcherrimus*) embryos exposed to TBT. Vertical bars represent the SD of the mean for three times. * Significantly different from the control by Fisher’s exact test (*p* < 0.05), ** *p* < 0.01.

**Figure 4 animals-13-03078-f004:**
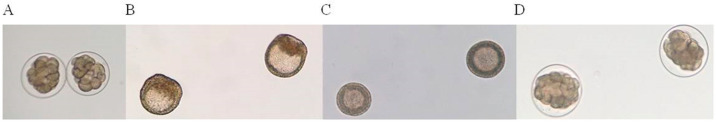
Development recovery test in sea urchin (*Hemicentrotus pulcherrimus*) embryos exposed to 4.87 ppb TBT. (**A**) morula-stage embryos were used 4 h after fertilization. (**B**) Gastrula stage embryos were present after 24 h of incubation following fertilization in the control medium. (**C**) Blastula stage embryos were present after 24 h of incubation following fertilization in media lacking 4.87 ppb TBT. (**D**) Morula stage embryos were present after 24 h of incubation following fertilization in media with 4.87 ppb TBT.

**Figure 5 animals-13-03078-f005:**
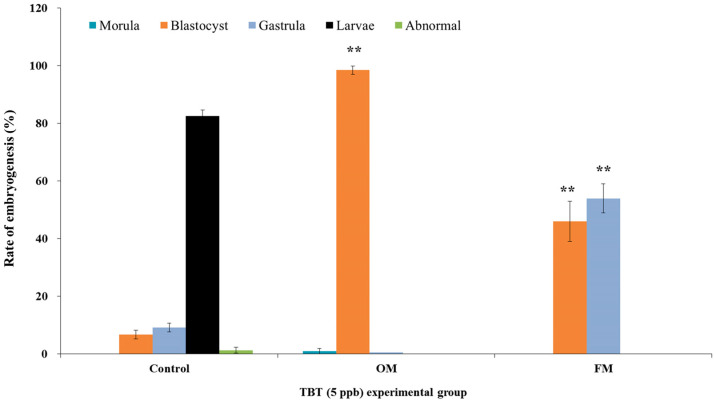
Graph showing the results of the development recovery test in sea urchin (*Hemicentrotus pulcherrimus*) embryos exposed to 4.73 ppb TBT for 64 h after fertilization. Control; Embryos and larvae were present after 64 h of incubation following fertilization with the original control medium. OM; Embryos were present after 64 h of incubation following fertilization in medium containing 4.73 ppb TBT. FM; Embryos were present after 64 h of incubation following fertilization in medium lacking 4.73 ppb TBT (** *p* < 0.01).

**Table 1 animals-13-03078-t001:** Experimental culture conditions using the sea urchin, *Hemicentrotus pulcherrimus*.

Test Parameters	Condition
Culture type	Static non-renewal 10 min—64 h toxicity test
Photoperiod	Ambient light condition and 8L:16D period
Temperature	16 ± 0.5 °C
pH	7.8–8.2
Salinity	32 ± 1.0 psu
Culture dish	6-well plate culture dish
Solution	Filtered (0.45 μm) and sterilized seawater
Solution exchange	None
Experiment period	10 min–64 h
Investigation item	Fertilization, larval development rates
Acceptability criterion	>90% fertilized eggs and pluteus larvae in control

## Data Availability

Not applicable.

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
