# Peer review of "Effects of Tributyltin-Contaminated Aquatic Environments and Remediated Water on Early Development of Sea Urchin (Hemisentrotus pulcherrimus)"

_animals, 2023, doi:10.3390/ani13193078_

Round 1

Reviewer 1 Report

Lines 32-33: The verb is missing

Line 48: Use another verb (ie."disrupt with affect");

Line 53: Why not also mention emerging PDBE pollutants known to be endocrine disruptors? Too old quotes;

Line 82: Add some examples of the state of the art on sea urchin and TBT and above all refer to more recent articles. The [26] reference dates from 1980. There are many more recent ones: Di Natale et al. 2019, Masullo et al., 2020 and Migliaccio 2015 refer to the effects of metals on embryo; Di Natale et al., 2022 explore the effects of PBDEs on embryos. There are many others exploring the effects of organic contaminants on embryos;

Line 108: why sperm has been washed once while eggs 3 times? Explain;

Fig 1: Why put "EGG" on the first image and not on the others? It would be enough to specify that they are all eggs in the legend of the figure;

Section 2.3: The p value symbol should be indicated with a small p in italics, not capital letters. Then, what software was used? (excel office package, Statistics, RStudio, etc.);

Line 173: Again the verb is missing;

Line 177: Figure 1 should be inserted here with the description of what has been obtained after treatment with TBT: in fact in figure 1 it is not pointed out that the shown abnormal egg is a result of TBT treatment;

Fig. 3: The ordinate axis is not % embryogensys (wrong written, by the way), but rate embryogenesis (%) or rate of normal development;

Line 232: Instead of writing "Fig 4" two times, why not introduce a “compared to control..”;

Lines 184-185: Why would this result indicate an acrosin deficiency? Specify better and give some concrete examples;

Line 277: Patents??

In general, the Results and Discussion part should be more thoroughly discussed. For example, the results of Figs 4 and 5 should be supported by the comparison with similar studies on sea urchin.

Conclusions are too simple and not punctual. The ecological significance of the results and future prospects do not emerge.

References are definitely too old.

Author Response

Thank you for your thorough review of our paper. 
We have diligently responded to the reviewer's comments provided and attached them in a file.

Reviewer 2 Report

The authors show an effects of TBT concentration on development of sea urchin. I feel that the topic and the results fit in animals and suitable for brief report.

This is a well-written paper containing interesting results that merit publication. For the benefit of the reader, however, there are several points that have to be clarified and improved before it can be published.

1. Lines 115, Why authors choose 30 minutes?

2. Lines 125, Why authors choose 16oC for the embryos culture?

3. Fig.2, Why the fertilization rate is relatively higher in 0.73 and 4.68. than those in 0.05, 0.09, 0.17, 0.43?

The English of the manuscript should be improved and edited by any native user, since there are several sentences with grammatical errors.

Author Response

(The authors gave the same response as above.)

Reviewer 3 Report

In the manuscript entitled “Effects of Tributyltin-contaminated aquatic environments and remediated water on the early development of sea urchin Hemisentrotus pulcherrimus” the authors examine the toxicity of tributyltin (TBT), an endocrine disruptor that is commonly used in industrial applications and is a common marine pollutant. In this report the authors examined the dose-dependent effects of TBT on fertilization and embryonic development of an Asian sea urchin species. The authors pre-treated sperm prior and then examined the fertilization rates, and then cultured embryos through early development in the absence or presence TBT. Lastly, the authors investigated the reversibility of TBT treatment by washing early embryos treated for 4 hours into fresh media.

TBT is a serious pollutant, and it is clear that this has been studied in other organisms, including other sea urchin species. While the results are straightforward, there are several issues that the authors need to correct prior to publication.

1.     Given that the authors cite multiple studies documenting the effects of TBT, including in sea urchins. In the Introduction, the authors should provide a stronger rationale for the study and in the results/discussion, provide some context for how their results fit into these published studies.

2.     What was the reasoning for only performing a pre-treatment of sperm and not eggs? The authors speculate that TBT may interfere with calcium signaling, yet don’t examine the potential effects on the egg’s ability to effect the slow block to polyspermy.

3.     There are several instances where the text needs a clearer explanation. The first sentence of the Results and Discussion is written in a manner that is misleading to the reader by describing the sperm pre-treatment and long-term treatments in a single sentence.

4.     The description of the washout experiment needs better phrasing both in the text and in the figure legend. The term FM is used in the figure, but RM is used in the legend, and neither term is used in the text. It is so poorly described that it is difficult to figure out what the results actually are.

Author Response

(The authors gave the same response as above.)

Round 2

Reviewer 3 Report

the authors have addressed the issues from both reviewers.